# MoMCE: Mixture of Modality and Cue Experts for Multimodal Deception Detection

## Abstract

Multimodal audio-visual deception detection aims to predict whether a person is lying by integrating visual and acoustic modalities, which has two main challenges: 1) the modality conflict problem and 2) heterogeneous cue representation difficulty. However, existing approaches 1) often overlook the differences across modalities for different individuals; and 2) typically rely on a single encoder to handle diverse and individual-specific cues, which limits models' representation capacity for heterogeneous cues. To address these challenges, we propose **MoMCE**, a novel model with mixture of modality and cue experts for deception detection. It consists of two key components: 1) **Prompt-aware Mixture of Modality Experts**, which employs a learnable prompt routing mechanism to generate adaptive instance-aware modality weight distributions for dynamic modality adjustment. In addition, we propose a consistency-aware expert weighting loss. For samples with high cross-modal consistency, it encourages balanced contributions across modalities. In contrast, for samples with strong conflicts, it reduces the entropy of the modality weight distribution to focus on more reliable modalities. 2) **Prompt-aware Mixture of Cue Experts**, which captures heterogeneous and diverse deceptive cues within each modality. This module introduces multiple experts with distinct semantic biases on top of a shared backbone to model different deceptive patterns. Additionally, we introduce a cue expert diversity loss to balance learning across multiple cue experts, promoting effective representation of diverse deceptive cues. Extensive experiments demonstrate that MoMCE adapts to variations in both cross-modal contributions and cue heterogeneity, achieving substantial improvements in deception detection performance.

## 1 Introduction

Deception is defined as intentional misleading DePaulo et al. (2003), with impacts on public safety Abouelenien et al. (2016b), judicial fairness Pérez-Rosas et al. (2015); Fornaciari & Poesio (2013), and economic stability Ding et al. (2019); Bajaj et al. (2023). Therefore, improving the accuracy of deception detection is of great importance for safeguarding judicial fairness and maintaining social stability. With the development of deep learning Vaswani et al. (2017); LeCun et al. (2015), non-intrusive multimodal analysis has gradually become an important direction in deception detection. Such approaches aim to automatically extract and model deception-related features from speech and visual signals, thereby enabling the identification of potential deceptive behaviors Karnati et al. (2021); Guo et al. (2023).

In multimodal deception detection tasks, it is often assumed that different modalities within the same video segment are consistent, meaning that a liar would reveal deceptive cues across all modalities. However, psychological studies DePaulo et al. (2003) indicate that because deceivers attempt to disguise themselves, they often find it difficult to maintain complete consistency across modalities. For example, an individual may have a natural expression, but their voice may reveal nervousness; conversely, their voice may remain steady, but their facial expressions may betray deception. This "modality conflict" phenomenon results in different representations of deception across modalities, with the importance of each modality varying across specific samples. Moreover, even within the same modality, deception cues exhibit significant inter-individual variability. For example, in the visual modality, some individuals display noticeable eye movements when lying, while others primarily reveal information through subtle facial expressions. In the audio modality, some individuals

speed up their speech or pause frequently, while others primarily express emotion through changes in voice tone. Such cue heterogeneity means that different individuals may rely on completely different behavioral patterns to convey deception cueswithin the same modality.

Existing methods primarily integrate multimodal information through common modality fusion techniques, such as feature concatenation Nam et al. (2023); Kang et al. (2024); Karnati et al. (2021), decision fusion Gupta et al. (2019); Wu et al. (2018), and adaptive fusion Ding et al. (2019); Guo et al. (2023); Zhang et al. (2022). These approaches typically apply the same fusion strategy to all samples, ignoring differences in the contribution of each modality across samples. As shown in Figure 1, the relative importance of different modalities can vary greatly across deceptive samples, and a uniform fusion strategy may even lead to contradictory predictions. On the other hand, to address deceptive cue heterogeneity, some studies attempt to introduce more handcrafted features Kang et al. (2024); Guo et al. (2024); Nam et al. (2023) or employ more powerful feature extractors Guo et al. (2023); Ji et al. (2025). However, these methods still rely on a single feature space and lack the ability to dynamically adapt to individual differences, thus limiting the model's generalization performance across individuals.

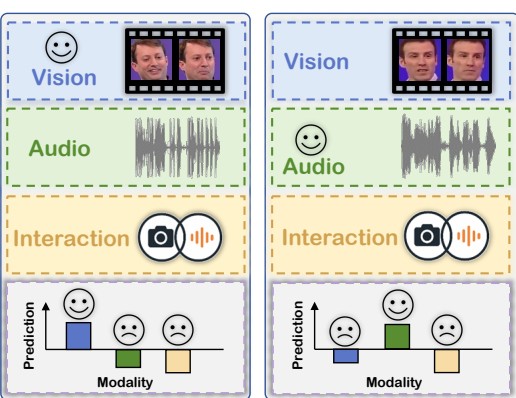

Figure 1: An example of modality importance in multimodal deception detection, where modality conflict can affect the prediction results.

In this paper, we propose a Mixture of Modality and Cue Experts (MoMCE) to jointly address the modality conflict and cue heterogeneity challenges. The model consists of two main components. First, to handle modality conflict, we design a prompt-aware mixture of modality experts. This module combines learnable prompt routing and gating mechanisms, treating each modality as an independent "expert." It adaptively assigns modality weights based on input features, thereby enabling dynamic modeling of modality contributions. Moreover, to mitigate performance degradation caused by incorrect routing, we introduce a consistency-aware expert weight loss: when the modalities are consistent (e.g., truthful samples or low-conflict deceptive samples), the weight distribution is constrained to remain balanced; when modality conflict is significant (high-conflict deceptive samples), the model is encouraged to reduce weight distribution entropy and focus on more reliable modalities, thus improving the robustness of inference.

Second, to address cue heterogeneity, we propose a novel model with prompt-aware mixture of cue experts. On top of a shared backbone encoder, we introduce multiple sets of prompts with different semantic biases to construct multiple cue-level experts, each specializing in capturing a different type of deceptive cue, thereby covering the diverse behavioral patterns of deception. In addition, during training, cue-level experts tend to overfit to the most salient cues, leading to insufficient learning of other potential cues Peng et al. (2022); Wang et al. (2024a;b); Wei et al. (2024). To address this, we design a cue diversity constraint that balances the learning process across cue experts, ensuring that various deception cues are effectively modeled. Our main contributions are as follows:

- To address modality conflicts, we propose a mixture of modality experts that dynamically generates sample-specific cross-modal weight distributions, enabling adaptive fusion of modality-level features for each sample. To further mitigate erroneous routing, we introduce a consistency-aware expert weighting loss to constrain the model's routing entropy and regulate the modality weights.

- We propose a mixture of cue experts that leverages prompts with different semantic priors to construct multiple cue-level experts for each modality, enabling the model to capture heterogeneous deception cues. To prevent expert collapse, we further design a cues diversity constraint that encourages experts to learn complementary and discriminative features.

- Extensive experiments on the DOLOS, Bag-of-Lies, and MU3D datasets show that our method outperforms state-of-the-art multi-modal deception detection approaches, demonstrating its effectiveness.

## 2 RELATED WORK

**Audio-Visual Deception Detection.**   Audio-visual joint learning for multimodal deception detection aims to identify deceptive behaviors by simultaneously modeling audio and visual modalities. Previous studies can be broadly categorized into two directions: optimizing modality fusion and enhancing deception cue representation. In the first category, early methods typically integrated hand-crafted multimodal features through simple feature concatenation or decision-level fusion Abouelenien et al. (2016a); Gupta et al. (2019); Rill-Garcia et al. (2019); Mathur & Matarić (2020b); Chebbi & Jebara (2023). With the rise of deep learning, some approaches began leveraging attention mechanisms to fuse deep representations, including CNN Gogate et al. (2017); Karnati et al. (2021), ResNet Ding et al. (2019), DNN Şen et al. (2020), LSTM Hsiao & Sun (2022), GNN Zhang et al. (2022), ViT Zhuo et al. (2024), and Wav2Vec Zhuo et al. (2024). More recently, research has shifted towards efficient and adaptive methods. For example, PECL Guo et al. (2023) introduces cross-modal adapters to learn latent alignments and relational features across modalities. However, such methods typically apply the same processing strategy to all input samples, ignoring potential modality conflicts. The second line of work Kang et al. (2024); Guo et al. (2024); Nam et al. (2023); Guo et al. (2023); Ji et al. (2025), attempts to capture richer deception cues by introducing more hand-crafted features or training more powerful encoders. However, these approaches still rely on a single feature space and lack the ability to dynamically adapt to individual differences. In contrast, our approach performs dynamic modality weighting and employs a mixture of cue experts to model heterogeneous cues, effectively addressing modality conflicts and cue heterogeneity, thereby enabling a more flexible and adaptive multimodal deception detection framework.

**Mixture-of-Experts.**   The core idea of a Mixture-of-Experts (MoE) model is to dynamically select the most suitable "experts" based on the input, where each expert specializes in capturing a specific type of feature or pattern. Through a collaborative mechanism among experts, the MoE effectively integrates the strengths of each expert while maintaining computational efficiency, thereby improving the model's predictive performance. In multimodal learning, recent studies have applied MoE structures to achieve several goals, including reducing computational cost Mustafa et al. (2022); Shen et al. (2023), handling modality missingness Yun et al. (2024), modeling modality interactions Cao et al. (2023); Akbari et al. (2023), dynamic modality fusion Cheng et al. (2024), cross-modal continual learning Huai et al. (2025), and aligning modality-specific and modality-invariant features Gao et al. (2024); Fang et al. (2025). Inspired by these advances, we decouple inter- and intra-modality processing, represent both modalities and deception cues as experts, and introduce prompt-aware structures to automatically aware expert competence and learn their importance weights. Thereby improving the ability to personalized model individual deceptive behaviors.

## 3 METHOD

### 3.1 OVERALL FRAMEWORK

Following typical multimodal deception detection approaches Guo et al. (2023), this paper aims to identify deception in videos using both visual and audio signals. Given a video segment $U$, the inputs consist of a visual modality ($v$) and an audio modality ($a$). The visual and audio streams are encoded into embedding sequences through a two-dimensional convolutional neural network (2D-CNN) and a one-dimensional convolutional neural network (1D-CNN), respectively. These two modalities are then projected into a unified embedding space:

$$X_v \in \mathbb{R}^{L \times D}, \quad X_a \in \mathbb{R}^{L \times D}, \tag{1}$$

where $L$ represents the sequence length and $D$ denotes the embedding dimension. Motivated by Ding et al. (2019), we model the interaction features as an independent modality and employ a cross-modal attention mechanism to align and integrate the visual and audio modalities, thereby obtaining the interaction embeddings.

Our overall framework is shown in Figure 2 and consists of two main parts. In the mixture of cue experts, each modality introduces multiple cue-level experts to capture diverse deception cues. These experts are then aggregated into modality-level features through a prompt-aware routing network.

$$Y_v = f_v(X_v), \quad Y_i = f_i(X_v, X_a), \quad Y_a = f_a(X_a), \tag{2}$$

Figure 2: **Framework of MoMCE.** It consists of two main stages. **In the MoCE stage**, each modality introduces multiple cue-level experts to capture different types of deceptive cues. These experts are then aggregated into modality-level features through a prompt-aware routing network. **In the MoME stage**, visual, audio, and interaction features are dynamically weighted via a modality-level prompt routing mechanism to assess the relative contribution of each modality.

where $f_v$, $f_i$, and $f_a$ denote the visual, interaction, and audio mixture of cue experts, respectively. In the mixture of modality experts stage, the visual features $Y_v$, audio features $Y_a$, and interaction features $Y_i$ are dynamically weighted by a modality-level prompt routing mechanism to evaluate the relative contribution of each modality. This produces a unified representation, which is then processed by a feed-forward network (FFN) to generate the final prediction.

## 3.2 PROMPT-AWARE MIXTURE OF CUE EXPERTS

To capture diverse deceptive cues within each modality, we propose a prompt-aware mixture of cue experts module. This module consists of multiple cue experts and a dynamic prompt routing network. Each cue expert is built upon a pretrained ViT Dosovitskiy et al. (2020) (for the visual modality) or W2V2 Baevski et al. (2020) (for the audio modality), augmented with learnable expert prompts to ensure both strong representational capacity and expert diversity. The dynamic prompt routing network utilizes learnable gating prompts to perceive the dynamic capabilities of experts to generate a cue-expert weight distribution, which is ultimately used to aggregate multiple cues. Unlike multimodal models in visual tasks Riquelme et al. (2021); Abbas & Andreopoulos (2020) and general multimodal models Li et al. (2025); Cheng et al. (2024), our approach leverages a prompt-aware dynamic routing mechanism to better model the complementarity and task-specific dependencies among experts. Formally, for visual or audio modalities, let the token sequence at the $n$-th Transformer block be

$$X^{(n)} = [x_1^{(n)}, \ldots, x_L^{(n)}]. \tag{3}$$

We prepend three types of prompts to the sequence: cross-modality gating prompts $p_{\text{cross}}^{(n,m)}$, intra-modality gating prompts $p_{\text{intra}}^{(n,m)}$, and expert prompts $p_{\text{exp}}^{(n,m)}$. The number of gating prompts is set to $T_{\text{g}}$, and the number of expert prompts is set to $T_{\text{e}}$. The input for the $m$-th expert at the $n$-th layer is

$$Z^{(n,m)} = [p_{\text{cross}}^{(1,n,m)}, \ldots, p_{\text{cross}}^{(T_{\text{g}},n,m)}, \; p_{\text{intra}}^{(1,n,m)}, \ldots, p_{\text{intra}}^{(T_{\text{g}},n,m)}, \; p_{\text{exp}}^{(1,n,m)}, \ldots, p_{\text{exp}}^{(T_{\text{e}},n,m)}, \; x_1^{(n)}, \ldots, x_L^{(n)}]. \tag{4}$$

Subsequently, the sequence is passed through the frozen, pre-trained Transformer block to compute feature correlations, yielding

$$Z^{(n,m)} = \text{TransformerBlock}\big(Z^{(n-1,m)}; \theta_{\text{shared}}^{(n)}\big), \tag{5}$$

**Prompt-aware Routing Network**. We extract $\bar{p}_{\text{intra}}^{(n,m)}$ from $Z^{(n,m)}$. This vector interacts with the expert prompts and the original tokens through a self-attention mechanism, enabling the network to perceive the response of each expert to the current modality features:

$$\bar{p}_{\text{intra}}^{(n,m)} = Z^{(n,m)}[\text{index of } p_{\text{intra}}^{(n,m)}]. \tag{6}$$

The routing score for each expert is then computed as

$$s^{(n,m)} = w^{n\top} \bar{p}_{\text{intra}}^{(n,m)}, \qquad \alpha^{(n,m)} = \frac{\exp\big(s^{n,m}\big)}{\sum_{j=1}^{M} \exp\big(s^{n,j}\big)}. \tag{7}$$

where $w^n$ is the learnable projection vector of modality $n$, $\alpha^{(n,m)}$ is the normalized routing weight obtained. we obtain $\tilde{Z}^{(n,m)}$ by removing the expert and intra-modality gating prompts from $Z^{(n,m)}$. The final fused feature of the MoCE module is obtained by

$$Y^n = \sum_{m=1}^{M} \alpha^{(n,m)} \cdot \tilde{Z}^{(n,m)}. \tag{8}$$

The visual and audio branches independently perform this computation, yielding $Y_v^n$ and $Y_a^n$. For the interaction branch, we compute the interactions information using $Y_v^n$ and $Y_a^n$. For the $m$-th interaction expert at the $n$-th layer, the inputs are

$$Z_a^{(n,m)} = [\,p_{a,g}^{(n,m)}, p_{a,e}^{(n,m)}, Y_a^n\,], \quad Z_v^{(n,m)} = [\,p_{v,g}^{(n,m)}, p_{v,e}^{(n,m)}, Y_v^n\,], \tag{9}$$

where $p_{a,g}^{(n,m)}$ and $p_{a,e}^{(n,m)}$ denote the learnable audio expert gating prompt and the audio expert prompt, respectively, and $p_{v,g}^{(n,m)}$ and $p_{v,e}^{(n,m)}$ are defined analogously. Within each interaction expert, cross-modal attention is applied to capture interactions between the two modalities:

$$\hat{Z}_a^{(n,m)} = \text{MHSA}(Z_a^{(n,m)}, Z_v^{(n,m)}, Z_v^{(n,m)}), \quad \hat{Z}_v^{(n,m)} = \text{MHSA}(Z_v^{(n,m)}, Z_a^{(n,m)}, Z_a^{(n,m)}), \tag{10}$$

where MHSA denotes the multi-head self-attention mechanism. After applying MHSA, we extract the gating prompts and token representations from $\hat{Z}_a^{(n,m)}$ and $\hat{Z}_v^{(n,m)}$ :

$$\overline{p}_{a,g}^{(n,m)}, \overline{Y}_a^{(n,m)} = \hat{Z}_a^{(n,m)}[\text{index of } p_{a,g}^{(n,m)}, Y_a^n], \quad \overline{p}_{v,g}^{(n,m)}, \overline{Y}_v^{(n,m)} = \hat{Z}_{v,g}^{(n,m)}[\text{index of } p_{v,g}^{(n,m)}, Y_v^n]. \tag{11}$$

We compute both the interaction features $\overline{Y}_i^{(n,m)}$ and the expert-level gating token $\overline{p}_{i,g}^{(n,m)}$ as

$$\overline{Y}_i^{(n,m)} = \phi\big(W_f[\overline{Y}_a^{(n,m)}, \overline{Y}_v^{(n,m)}]\big), \quad \overline{p}_{i,g}^{(n,m)} = \phi\big(W_c[\overline{p}_{a,g}^{(n,m)}, \overline{p}_{v,g}^{(n,m)}]\big), \tag{12}$$

where $[\cdot,\cdot]$ denotes concatenation, $W_f$ and $W_c$ are trainable projection matrices, and $\phi(\cdot)$ is the ReLU activation. Finally, we apply Equations (7)–(8) to perform expert selection and weighted fusion across the $M$ experts, yielding the final interaction representation:

$$Y_i^n = \text{MoE-Fusion}(\overline{Y}_i^{(n,m)}, \overline{p}_{i,g}^{(n,m)}). \tag{13}$$

**Cue Expert Diversity Loss.** The *Mixture of Cue Experts* aim to capture heterogeneous deceptive cues. However, during the training of multiple experts, a "dominant cue" phenomenon often arises: several experts tend to repeatedly focus on the most salient and easy-to-learn deceptive cues, while other potential cues receive insufficient attention. To mitigate this issue, we introduce the *Cue Expert Diversity Loss* to balance the learning among experts. The loss is formulated as follows:

$$\mathcal{L}_{\text{div}} = \frac{1}{BM(M-1)} \sum_{b=1}^{B} \sum_{i \neq j} \cos(\mathbf{R}_{b,i}, \mathbf{R}_{b,j}), \tag{14}$$

where $\mathbf{R} \in \mathbb{R}^{B \times M \times D}$ denotes the expert representations for a batch of $B$ samples. Here, $\cos(\mathbf{R}_{b,i}, \mathbf{R}_{b,j})$ denotes the cosine similarity between the $i$-th and $j$-th expert representations of the $b$-th sample, $M$ denotes the number of experts. This encourages expert to provide complementary features, thereby reducing redundancy and improving the capture of diverse deceptive cues.

### 3.3 Prompt-aware Mixture of Modality Experts

In the MoME, we consider three types of modality experts: visual $Y_{\text{vis}}$, audio $Y_{\text{aud}}$ and interaction $Y_{\text{inter}}$. After being encoded by the MoCE, three modalities include cross-modality gating prompts and feature tokens, each cross-modality gating prompt has perceived the representation of modality. Therefore, we use the cross-modal gating prompts to generate the experts' weight distributions. Specifically, we extract the cross-modality gating prompts $\overline{p}_{\text{cross}}^{(m)}$ and feature tokens $\overline{X}_k$:

$$\overline{p}_{\text{cross}}^{(m)}, \overline{X}_k = Y_k[\text{index of } p_{\text{cross}}^{(m)}, X], \quad k \in \{\text{vis}, \text{aud}, \text{inter}\}. \tag{15}$$

Afterwards, we compute cross-modal scores and softmax weights from these descriptors:

$$u_k = \mathbf{w}_{\text{cross}}^{\top} \overline{p}_{\text{cross}}^{(m)}, \qquad \beta_k = \frac{\exp(u_k)}{\sum_{j \in \{\text{vis,aud,inter}\}} \exp(u_j)}. \tag{16}$$

where $\mathbf{w}_{\mathrm{cross}}$ is a learnable weight, $u_k$ is the score for modality $k$, $\beta_k$ is the softmax weight for modality $k$. Finally, we obtain the fused multimodal representation $Y_{\mathrm{c}}$ by weighting:

$$Y_{\mathrm{c}} = \sum_{k \in \{\mathrm{vis,aud,inter}\}} \beta_k \overline{X}_k. \tag{17}$$

Based on this fused representation, we apply a feed-forward network (FFN) to obtain the prediction $\hat{y}$, and compute the loss function by comparing it with the ground-truth label $y$.

**Consistency-aware Expert Weight Loss.** Although the MoME can assign expert weights according to each expert's capability, due to the highly individualized nature of modality conflicts across different samples, the learned modality weight distribution may still rely on unreliable modalities. We introduce a consistency-aware expert weight loss, which constrains the modality weight distribution based on the conflict level. Specifically, the modality conflict $\kappa$ is defined as:

$$\kappa = 1 - \frac{1}{3 * (3 - 1)} \sum_{k \neq l} \cos(\overline{X}_k, \overline{X}_l), \tag{18}$$

where $k, l \in \{\mathrm{vis, aud, inter}\}$ index the visual, audio, and interaction, and $\cos(\overline{X}_k, \overline{X}_l)$ computes the cosine similarity between the mean representations of modality $k$ and $l$. Given a threshold $c_{\mathrm{th}}$, we compare the samples' conflict scores $\kappa$ with the threshold and divide them into three categories: *genuine samples*, *low-conflict deceptive samples*, and *high-conflict deceptive samples*. For genuine and low-conflict deceptive samples, we adopt a more *uniform* weight distribution in the loss to encourage modeling modality diversity, while for high-conflict deceptive samples, we promote a more *concentrated* weight distribution to emphasize the most reliable modality. We measure the dispersion of the weights using the routing entropy Fang et al. (2025), defined as:

$$\mathrm{H}(\beta^{(b)}) = - \sum_{k} \beta_k^{(b)} \log(\beta_k^{(b)} + \epsilon), \tag{19}$$

where $\epsilon$ is a small constant for numerical stability. The consistency-aware expert weight loss $\mathcal{L}_{\mathrm{CEW}}$ is defined as:

$$\mathcal{L}_{\mathrm{CEW}} = \frac{1}{B} \sum_{b=1}^{B} \left( \mathrm{H}(\beta^{(b)}) - \mathrm{H}_b^* \right)^2 \tag{20}$$

where $\mathrm{H}^*$ denotes the target entropy. For genuine and low-conflict deceptive samples, we set $\mathrm{H}^* = 1.0$, while for high-conflict deceptive samples, we set $\mathrm{H}^* = 0.0$.

### 3.4 LOSS FUNCTIONS

We combine the above loss functions to obtain the overall training objective:

$$\mathcal{L} = \mathcal{L}_{\mathrm{cls}} + \lambda_{\mathrm{aux}} \mathcal{L}_{\mathrm{CEW}} + \lambda_{\mathrm{div}} \mathcal{L}_{\mathrm{div}}, \tag{21}$$

where $\mathcal{L}_{\mathrm{cls}}$ denotes cross-entropy loss. $\lambda_{\mathrm{aux}}$ and $\lambda_{\mathrm{div}}$ are hyperparameters that control the relative importance of the auxiliary losses. The overall workflow of MoMCE is detailed in Appendix A.1.

## 4 EXPERIMENTAL ANALYSIS

### 4.1 IMPLEMENTATION DETAILS

For the visual modality, we use AlphaPose Fang et al. (2017) to extract faces from videos. For each face clip, we uniformly sample 64 frames, normalize them, and resize them to $160 \times 160$ pixels. The face images are encoded using a 2D-CNN module to produce 256-dimensional features. For the audio data, the raw speech signal is resampled and encoded using a 1D-CNN module, resulting in a 512-dimensional feature vector for each audio sample. The visual and audio tokens are then projected into 768 dimensions through a linear projection layer. We train the model for 100 epochs using the Adam optimizer with a learning rate of $1 \times 10^{-5}$. The consistency-aware expert weight loss is inactive for the first 30 epochs to avoid early interference from unstable modality features. Model performance is evaluated in terms of Accuracy (ACC), F1-score (F1), and Area Under the Curve (AUC). All training and inference are conducted on two H100 GPUs. Unless otherwise specified, the number of experts $M$ in MoCE is set to 8, with $T_{\mathrm{g}} = 1$ gating prompt and $T_{\mathrm{e}} = 4$ expert prompts. The hyperparameters $\lambda_{\mathrm{div}}$ and $\lambda_{\mathrm{aux}}$ are both set to 0.1, and the threshold is set to $c_{\mathrm{th}} = 0.5$. The analysis of the hyperparameters $M$, $T_{\mathrm{g}}$, $T_{\mathrm{e}}$ and $c_{\mathrm{th}}$ is presented in Appendix A.3.

Table 1: In-domain evaluation results. **Bold** numbers indicate the best results.

| Method | ACC | F1 | AUC | Method | ACC | F1 | AUC |
|---|---|---|---|---|---|---|---|
| SVM (OpenFace) | 53.55 | 69.75 | 54.30 | Random Forest (LBP) | 55.26 | – | – |
| DT (OpenFace) | 50.58 | 53.58 | 50.58 | MLP (LBP) | 49.90 | – | – |
| RF (OpenFace) | 53.67 | 61.75 | 54.66 | SVM (LBP) | 53.25 | – | – |
| AdaBoost (OpenFace) | 50.57 | 55.36 | 50.35 | LieNet (Face) | 55.65 | 30.30 | 51.07 |
| SVM (AU) | 52.76 | 68.13 | 52.42 | FacialCueNet (Face) | 56.23 | **63.26** | 59.53 |
| DT (AU) | 51.73 | 54.53 | 51.73 | DDABG (Face) | 56.66 | 55.17 | 57.89 |
| RF (AU) | 50.45 | 58.08 | 51.57 | PECL (Face) | 58.94 | 43.06 | 58.44 |
| AdaBoost (AU) | 48.76 | 52.95 | 47.35 | | | | |
| LSTM (OpenFace) | 56.28 | 59.28 | 58.54 | KNN (MFCC) | 56.22 | – | – |
| LSTM (AU) | 56.46 | 63.43 | 58.68 | LieNet (Audio) | 56.15 | 24.38 | 52.82 |
| LSTM (RN18) | 59.72 | 64.15 | 56.68 | PECL (Audio) | 58.43 | 56.70 | 57.29 |
| LieNet (Face) | 59.67 | 73.24 | 53.14 | | | | |
| FacialCueNet (Face) | 60.98 | 68.65 | 61.99 | LieNet (Face+Audio) | 59.78 | 58.14 | 58.09 |
| DDABG (Face) | 55.47 | 62.52 | 52.13 | PECL (Face+Audio) | 59.51 | 51.06 | 59.41 |
| PECL (Face) | 61.44 | 69.42 | 58.89 | **MoMCE (Face+Audio)** | **61.09** | 52.91 | **60.93** |
| AFFAKT (Face) | 67.64 | 70.54 | 72.12 | | | | |

<table>
<tr><td colspan="8" align="center">(b) Results on Bag-of-Lies</td></tr>
</table>

| Method | ACC | F1 | AUC |
|---|---|---|---|
| MLP (MFCC) | 58.10 | 59.63 | 61.34 |
| MLP (OpenSMILE) | 55.37 | 68.67 | 53.25 |
| MLP (W2V2) | 54.21 | 43.83 | 53.69 |
| LieNet (Audio) | 57.14 | 72.72 | 50.00 |
| PECL (Audio) | 59.19 | 73.46 | 52.54 |
| AFFAKT (Audio) | 61.98 | 68.22 | 63.91 |
| LieNet (Face+Audio) | 56.50 | 69.72 | 51.02 |
| PECL (Face+Audio) | 64.75 | 71.20 | 62.71 |
| AFFAKT (Face+Audio) | 68.10 | 70.73 | 72.26 |
| **MoMCE (Face+Audio)** | **79.10** | **82.38** | **78.09** |

| Method | ACC | F1 | AUC |
|---|---|---|---|
| LieNet (Face) | 57.19 | 57.62 | 57.19 |
| FacialCueNet (Face) | 57.64 | 59.13 | 50.89 |
| DDABG (Face) | 55.63 | 51.99 | 52.20 |
| PECL (Face) | 54.68 | 47.19 | 54.64 |
| LieNet(Audio) | 53.73 | 44.52 | 53.73 |
| PECL(Audio) | 56.25 | 45.72 | 56.10 |
| LieNet (Face+Audio) | 53.48 | 33.62 | 54.75 |
| PECL (Face+Audio) | 55.31 | 60.07 | 55.31 |
| **MoMCE (Face+Audio)** | **58.58** | **60.35** | **58.69** |

(a) Results on the DOLOS dataset          (c) Results on MU3D

## 4.2 DATASET AND PROTOCOL

We evaluate the performance of MoMCE on three datasets: DOLOS Guo et al. (2023), Bag-of-Lies Gupta et al. (2019), and MU3D Lloyd et al. (2019). These datasets cover a variety of scenarios, including game shows, spontaneous lies, and real-world simulations, allowing for comprehensive evaluation of the model under diverse conditions. For the DOLOS dataset, we follow the three official train-test splits provided by the dataset and report the average results. For the Bag-of-Lies dataset, we use the B group containing all video samples. In addition, we follow the official participant-based split Gupta et al. (2019) and perform three-fold cross-validation. For the MU3D dataset, to eliminate the influence of identity on the experimental results, we divide the 80 participants into four identity groups, each containing 20 participants. In the experiments, we perform four-fold cross-validation, and the final results are reported as the average of the four experiments. All compared methods within the same dataset follow the same evaluation protocol. Detailed dataset descriptions and example visualizations are provided in the Appendix A.2.

## 4.3 IN-DOMAIN EVALUATION

Under the same dataset settings, we compare MoMCE with state-of-the-art automatic deception detection (ADD) methods, with the results on DOLOS, Bag-of-Lies, and MU3D shown in Tables 1a, 1b, and 1c, respectively. Following the practice of Guo et al. (2023) and Ji et al. (2025), we select traditional machine learning methods and deep learning models as baselines and conduct comparisons from visual, audio, and audiovisual perspectives. For traditional methods, visual features include OpenFace features, Action Units (AU) features, and LBP features, while acoustic features include MFCC (Mel-Frequency Cepstral Coefficients) and OpenSMILE features. In visual classification tasks, commonly used classifiers such as Support Vector Machines (SVM), Decision Trees (DT), and Random Forests (RF) are employed, whereas for acoustic classification tasks, Mul-

Table 2: Cross-dataset results on DOLOS (D), Bag-of-lies (B), and MU3D (M); "X&Y→Z" denotes training on X and Y, testing on Z.

| Method | M&B to D | | | D&M to M | | | D&B to B | | | Average | | |
|---|---|---|---|---|---|---|---|---|---|---|---|---|
| | ACC | F1 | AUC | ACC | F1 | AUC | ACC | F1 | AUC | ACC | F1 | AUC |
| LieNet (V) | 53.42 | 69.62 | 49.84 | 52.19 | 62.77 | 52.19 | 55.38 | 42.23 | 55.32 | 53.66 | 58.21 | 52.45 |
| FacialCueNet (V) | 48.84 | 31.68 | 50.63 | 52.56 | 48.97 | 51.26 | 57.10 | 61.28 | 55.99 | 52.84 | 47.31 | 52.63 |
| PECL (V) | 54.01 | 70.09 | 50.08 | 51.25 | 50.63 | 51.25 | 54.46 | 42.19 | 54.40 | 53.24 | 54.30 | 51.91 |
| LieNet (A) | 53.91 | 63.50 | **52.29** | 53.75 | 36.21 | 53.75 | 52.92 | 49.17 | 52.90 | 53.53 | 49.63 | 52.98 |
| PECL (A) | 54.63 | 68.80 | 51.36 | 51.56 | **67.09** | 51.56 | 57.54 | **63.68** | 57.59 | 54.58 | **66.52** | 53.50 |
| LieNet (V+A) | 54.40 | 68.23 | 51.53 | 54.69 | 50.51 | 54.69 | 51.08 | 59.75 | 51.14 | 53.39 | 59.50 | 52.45 |
| PECL (V+A) | 54.51 | 69.55 | 50.92 | 51.25 | 66.95 | 51.25 | 55.38 | 52.46 | 55.37 | 53.71 | 62.99 | 52.51 |
| MoMCE (V+A) | **57.80** | **72.88** | 50.71 | **56.25** | 60.49 | **55.26** | **59.20** | 56.68 | **59.17** | **57.75** | 63.35 | **55.05** |

Table 3: Ablations on MoME (Mixture of Modality Experts) and MoCE (Mixture of Cue Experts).

| MoME | MoCE | DOLOS | | | BOL | | | MU3D | | |
|---|---|---|---|---|---|---|---|---|---|---|
| | | ACC | F1 | AUC | ACC | F1 | AUC | ACC | F1 | AUC |
| ✓ | | 61.73 | 69.85 | 58.97 | 59.87 | 41.94 | 58.30 | 57.50 | 58.35 | 57.50 |
| | ✓ | 65.62 | 71.65 | 63.86 | 60.40 | 44.14 | 59.44 | 57.63 | 54.64 | 53.64 |
| ✓ | ✓ | **79.10** | **82.38** | **78.09** | **61.09** | **52.91** | **60.93** | **58.58** | **60.35** | **58.69** |

tilayer Perceptrons (MLP) and KNN classifiers are adopted ( Mathur & Matarić (2020a); Avola et al. (2019); Yang et al. (2021); Gupta et al. (2019). Deep learning baselines include ResNet18 (RN18) + LSTM Ding et al. (2019), W2V2 + MLP Gogate et al. (2017), as well as recent ADD models such as LieNet Karnati et al. (2021), FacialCueNet Nam et al. (2023), DDABG Kang et al. (2024), PECL Guo et al. (2023), and AFFAKT Ji et al. (2025).

**Results and Analysis.** As shown in Table 1, MoMCE achieves significant improvements on accuracy (ACC) over existing methods: on the DOLOS dataset, ACC increases by 11% compared to AFFAKT Ji et al. (2025) (68.10%→79.10%), and on the Bag-of-Lies dataset by 1.31% (59.78%→61.09%). Similar gains are observed in AUC, e.g., an improvement of 5.83% on DOLOS (72.26%→78.09%). These improvements are consistent across Bag-of-Lies and MU3D. It is worth noting that our method yields slightly lower F1 than FacialCueNet Nam et al. (2023), mainly because FacialCueNet leverages multiple handcrafted facial deception features to boost recall. However, handcrafted features often have limited generalization in complex scenarios, which explains why FacialCueNet underperforms in ACC and AUC. From a methodological perspective, MoMCE consistently outperforms other ADD approaches. Compared with recent audiovisual learning methods such as PECL and AFFAKT, MoMCE dynamically estimates the importance of each modality per sample and performs adaptive fusion accordingly. Furthermore, the multi-expert design within each modality encourages multi-cue learning, enhancing intra-modal representation of deception cues and achieving joint optimization across and within modalities.

### 4.4 CROSS-DOMAIN EVALUATION

To further assess the effectiveness of **MoMCE**, we conduct *cross-domain evaluation* in Table 2. Across three datasets and three metrics (12 metrics in total), **MoMCE** achieves the best performance on 8 metrics. Notably, the average precision is improved from 54.58% (PECL(A)) to 57.75%. Compared to the *in-domain setting*, the *cross-domain setting* is considerably more challenging due to domain shifts. For instance, DOLOS consists of drama scenes, whereas Bag-of-lies and MU3D mainly contain laboratory scenes, leading to different levels of cue exposure. Despite these challenges, **MoMCE** consistently outperforms all competing methods, demonstrating its strong generalization ability.

### 4.5 ABLATION STUDY AND ANALYSIS

**Effectiveness of MoME and MoCE.** We evaluate the key components of MoMCE, including MoME and MoCE, on three datasets, as shown in Table 3. MoME yields a significant performance improvement. This demonstrates that dynamically fusing modality features according

Table 4: Ablations on CEDL and CEWL loss functions.

| Methods | DOLOS | | | BOL | | | MU3D | | |
|---|---|---|---|---|---|---|---|---|---|
| | ACC | F1 | AUC | ACC | F1 | AUC | ACC | F1 | AUC |
| Ours (w/o CEDL) | 75.65 | 79.62 | 74.44 | 60.09 | 33.81 | 57.02 | 57.81 | 43.01 | 57.35 |
| Ours (w/o CEWL) | 76.52 | 79.55 | 76.00 | 60.56 | 46.87 | 57.95 | 58.44 | 49.26 | 58.44 |
| Ours | **79.10** | **82.38** | **78.09** | **61.09** | **52.91** | **60.95** | **58.58** | **60.35** | **58.69** |

Figure 3: t-SNE visualization of feature distribution on DOLOS.

to sample-level importance enables the model to leverage complementary information from each modality more effectively, leading to better multi-modal representation learning. Furthermore, we observe that integrating MoCE also leads to a significant performance improvement. This result highlights the necessity of modeling heterogeneous cues within each modality, and shows that MoCE can capture multiple informative cues, filling the gap left by a single-encoder design.

**Effectiveness of CEDL and CEWL.** We further conduct ablation studies on CEDL and CEWL, as reported in Table 4. Both losses consistently improve performance, suggesting that without additional regularization, the model tends to over-focus on homogeneous cues and ignore other discriminative

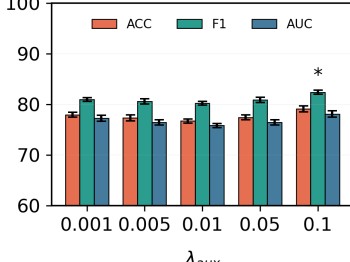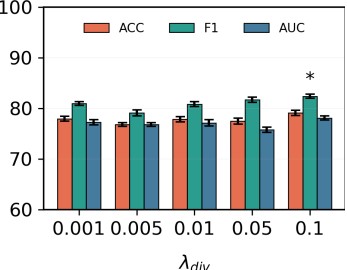

Figure 4: Effect of $\lambda_{aux}$ and $\lambda_{div}$ on ACC, F1, and AUC (with std. bars); asterisks indicate significant improvements.

information. CEDL explicitly encourages the model to learn diverse cues, while CEWL mitigates the modality conflict issue by enforcing balanced learning on conflict-prone samples. In addition, we vary the weights of CEDL and CEWL, as shown in Figure 4, and observe that the performance generally increases with higher weights, further confirming their effectiveness.

**Visualization.** We also visualize the feature distributions of MoMCE, MoMCE without MOCE, and MoMCE without MOME in Figure 3, providing a qualitative comparison. We conduct the experiment on the test set of Protocol 1 from the DOLOS dataset, projecting sample features into a two-dimensional space using t-SNE. The results show that the feature distributions of MoMCE without MOCE and MoMCE without MOME are irregular, while the features of MoMCE are more compact and consistent, with clearer separability among classes. These results indicate that MoMCE enhances feature discriminability, leading to more accurate deception detection.

## 5 CONCLUSION

In this paper, we address the challenges of modality conflicts and cues heterogeneity in audiovisual deception detection. We propose a Mixture of Modality and Cue Experts, MoMCE, which consists of two key components: a prompt-aware mixture of modality experts that adaptively adjusts modality contributions with consistency-aware expert weight regularization, and a prompt-aware mixture of cue experts that captures diverse and heterogeneous cues within each modality with cue diversity regularization. Extensive experiments on multiple multimodal deception detection benchmarks demonstrate that MoMCE achieves superior accuracy and robustness compared to existing methods.

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

# A APPENDIX

## A.1 OVERALL WORKFLOW OF MoMCE

We summarize the overall workflow of our *Mixture of Modality and Cue Experts* (MoMCE) framework, which is presented in Algorithm 1. Given a video segment $U$, both visual and audio modalities are first encoded into token sequences. Each modality is then processed by a *Mixture of Cue Experts* (MoCE) module to capture diverse intra-modal deception cues. In this stage, multiple experts are trained in parallel, and their outputs are dynamically weighted via a prompt-aware routing mechanism to form modality-level representations.

Following this, the *Mixture of Modality Experts* (MoME) stage fuses the visual, audio, and cross-modal interaction features. Modality-specific gating prompts are used to compute adaptive weights, allowing the model to adjust the relative contribution of each modality according to the input sample. This design mitigates the effect of modality conflicts, which are common in deception detection tasks.

Finally, the fused multimodal representation is passed through a feed-forward network to generate the final prediction. The overall training objective combines the main classification loss with auxiliary losses that encourage both expert diversity and consistency-aware gating, as described in Algorithm 1. This structured workflow allows MoMCE to effectively capture both fine-grained intra-modal deceptive cues, while adaptively balancing the contribution of each modality.

## A.2 DETAILED DATASET DESCRIPTIONS AND EXAMPLE VISUALIZATIONS

We evaluate the performance of MoMCE on three datasets: DOLOS Guo et al. (2023), Bag-of-Lies Gupta et al. (2019), and MU3D Lloyd et al. (2019). These datasets cover a variety of scenarios, including game shows, spontaneous lies, and real-world simulations, allowing for a comprehensive evaluation of the model under diverse conditions. Figure 6 shows the illustration of each dataset. The DOLOS dataset is sourced from a television game show and contains 1,675 video samples, including 899 deceptive videos and 776 truthful videos, recorded by 213 different participants. In our experiments, we follow the three official train-test splits provided by the dataset and report the averaged results. The Bag-of-Lies dataset is designed for spontaneous deception detection and contains 325 videos, with 162 deceptive videos and 163 truthful videos. The dataset is recorded by 35 participants, each providing both truthful and deceptive samples. The dataset is divided into two subsets: Group A and Group B. Group B contains all video samples, while Group A only includes samples recorded with intrusive physiological sensors. Since our method relies on non-intrusive video analysis, we perform evaluations on the larger Group B. Previous studies on Bag-of-Lies (e.g., Karnati et al. (2021); Gupta et al. (2019); Nam et al. (2023); Kang et al. (2024)) use inconsistent train-test splits, making direct comparisons difficult. To ensure comparability, we follow the official participant-based split Gupta et al. (2019), conduct three-fold cross-validation, and reproduce the results of Karnati et al. (2021); Nam et al. (2023); Kang et al. (2024)) under the same settings. The MU3D dataset contains 320 videos recorded by 80 different subjects, with each subject recording four videos, two truthful and two deceptive. To eliminate the influence of identity on experimental results, we divide the 80 subjects into four identity groups, each containing 20 subjects and 80 videos. In cross-validation, we use three groups for training and the remaining group for testing, repeating this four times so that each group serves as the test set once. We report the final results as the averages over the four experiments.

## A.3 HYPER-PARAMETER ANALYSIS.

To validate the robustness of MoMCE, we conduct a sensitivity analysis on its hyperparameters. Since the focus is on the overall performance of the model, we test various values of the threshold $c_{th}$, the number of intra-modality experts $M$, the number of expert prompts $T_e$, and the number of gating prompts $T_g$ on the three dataset. Specifically, the sensitivity analysis is performed by varying one target hyperparameter while keeping the others fixed at their baseline values used in the experiments. Figure 5 shows the average results of different hyperparameter settings across the datasets. The results indicate that the overall evaluation metrics remain relatively stable, suggesting that the performance of the proposed method is not sensitive to the values of these hyperparameters.

---

**Algorithm 1** Prompt-aware Mixture of Modality and Cue Experts (MoMCE)

---

**Require:** Video segment $U$ with visual $v$ and audio $a$, number of cue experts $M$, Transformer layers $N$, prompt counts $T_{\text{g}}, T_{\text{e}}$
**Ensure:** Final prediction $\hat{y}$

1: **Step 1: Encode visual and audio modalities**
2: $X_v \leftarrow \text{2D-CNN}(v)$
3: $X_a \leftarrow \text{1D-CNN}(a)$
4: **Step 2: Mixture of Cue Experts (MoCE) for each modality**
5: **for** modality $k \in \{\text{vis}, \text{aud}\}$ **do**
6:     **for** expert $m = 1$ to $M$ **do**
7:         Prepend prompts:
8:         $Z_0^{(n,m)} = [p_{\text{cross}}^{(1:T_{\text{g}},n,m)}, p_{\text{intra}}^{(1:T_{\text{g}},n,m)}, p_{\text{exp}}^{(1:T_{\text{e}},n,m)}, X_k]$
9:         **for** layer $n = 1$ to $N$ **do**
10:             $Z_n^{(m)} \leftarrow \text{TransformerBlock}(Z_{n-1}^{(m)}; \theta_{\text{shared}}^{(n)})$
11:         **end for**
12:         Extract intra-modality gating prompt: $\overline{p}_{\text{intra}}^{(n,m)} \leftarrow Z_N^{(m)}[\text{index of } p_{\text{intra}}^{(n,m)}]$
13:         Compute routing score: $s^{(n,m)} = w^{n\top} \overline{p}_{\text{intra}}^{(n,m)}$
14:     **end for**
15:     Softmax over experts: $\alpha^{(n,m)} = \text{softmax}([s^{(n,1)}, \dots, s^{(n,M)}])$
16:     Fuse expert outputs: $Y_k^n = \sum_{m=1}^{M} \alpha^{(n,m)} \cdot \tilde{Z}_N^{(m)}$
17: **end for**
18: **Step 3: Interaction Experts**
19: **for** interaction expert $m = 1$ to $M$ **do**
20:     $Z_{\text{aud}}^{(n,m)} = [p_{\text{a,g}}^{(n,m)}, p_{\text{a,exp}}^{(n,m)}, Y_{\text{aud}}^n]$
21:     $Z_{\text{vis}}^{(n,m)} = [p_{\text{v,g}}^{(n,m)}, p_{\text{v,exp}}^{(n,m)}, Y_{\text{vis}}^n]$
22:     Apply cross-modal attention (MHSA) to obtain $\hat{Z}_{\text{aud}}^{(n,m)}, \hat{Z}_{\text{vis}}^{(n,m)}$
23:     Extract features: $\overline{Y}_{\text{aud}}^{(n,m)}, \overline{p}_{\text{a,g}}^{(n,m)}, \overline{Y}_{\text{vis}}^{(n,m)}, \overline{p}_{\text{v,g}}^{(n,m)}$
24:     Compute interaction representations:
25:     $\overline{Y}_{\text{i}}^{(n,m)} = \phi(W_f[\overline{Y}_{\text{aud}}^{(n,m)}, \overline{Y}_{\text{vis}}^{(n,m)}])$
26:     Compute gating token: $\overline{p}_{\text{i,g}}^{(n,m)} = \phi(W_c[\overline{p}_{\text{a,g}}^{(n,m)}, \overline{p}_{\text{v,g}}^{(n,m)}])$
27: **end for**
28: Fuse interaction experts via MoE: $Y_{\text{inter}}^n = \text{MoE-Fusion}(\{\overline{Y}_{\text{i}}^{(n,m)}, \overline{p}_{\text{i,g}}^{(n,m)}\}_{m=1}^{M})$
29: **Step 4: Mixture of Modality Experts (MoME)**
30: **for** modality $k \in \{\text{vis}, \text{aud}, \text{inter}\}$ **do**
31:     Extract cross-modal gating prompt: $\overline{p}_{\text{cross}}^{(m)}, \overline{X}_k \leftarrow Y_k[\text{index of } p_{\text{cross}}^{(m)}, X_k]$
32:     Compute cross-modal score: $u_k = \mathbf{w}_{\text{cross}}^{\top} \overline{p}_{\text{cross}}^{(m)}$
33: **end for**
34: Softmax over modalities: $\beta_k = \text{softmax}([u_{\text{vis}}, u_{\text{aud}}, u_{\text{inter}}])$
35: Fuse modalities: $Y_{\text{c}} = \sum_k \beta_k \overline{X}_k$
36: **Step 5: Prediction and loss**
37: $\hat{y} = \text{FFN}(Y_{\text{c}})$
38: Compute total loss: $\mathcal{L} = \mathcal{L}_{\text{cls}} + \lambda_{\text{aux}} \mathcal{L}_{\text{CEW}} + \lambda_{\text{div}} \mathcal{L}_{\text{div}}$

---

Figure 5: Effects of varying $c_{\text{th}}$, $M$, $T_{\text{e}}$ and $T_{\text{g}}$ on model performance. ACC, F1, and AUC are plotted to illustrate trends and relative stability across hyperparameter settings.

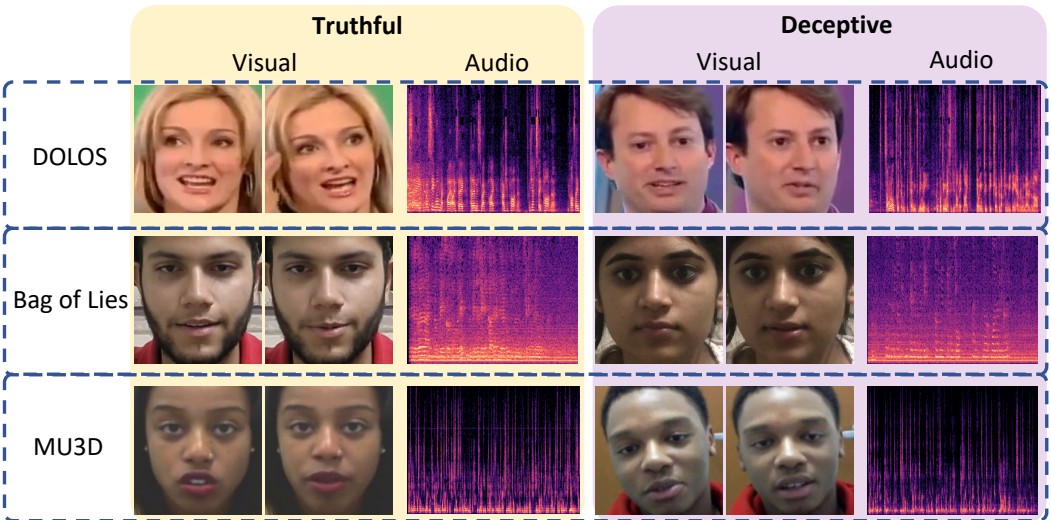

Figure 6: Visualization of sample examples from the DOLOS, Bag-of-Lies, and MU3D datasets.

