# OpenReview forum: "MoMCE: Mixture of Modality and Cue Experts for Multimodal Deception Detection"
_ICLR.cc/2026/Conference — ICLR 2026 Conference Withdrawn Submission_

### Official Review · Reviewer_9JSR · 2025-10-28

**Soundness:** 2
**Presentation:** 3
**Contribution:** 2
**Rating:** 4
**Confidence:** 5

**Summary:**

The paper proposes MoMCE, a model for multimodal audio-visual deception detection, addressing challenges of modality conflict and heterogeneous cue representation. It introduces prompt-aware Mixture of Modality Experts to adaptively weight modalities per sample and prompt-aware Mixture of Cue Experts to capture diverse, individual-specific deceptive cues. Specialized losses encourage balanced contributions across modalities and diversity among cue experts. Experiments show that MoMCE effectively handles both cross-modal variations and cue heterogeneity, achieving noticeable performance improvements over compared methods.

**Strengths:**

1) The paper does not invent mixture-of-experts or prompts (they exist in other domains) but it combines these techniques in a unique, way tailored to multimodal deception detection. This exact combination of adaptive MoE for both modalities and cues with specialized losses has likely not been done before in this task.

2) The study tackles important challenges in multimodal learning: modality conflict and heterogeneous cue representation. Given the increasing use of prompt-based and adaptive modeling approaches, the paper provides a modern solution to problems that have remained largely unresolved in prior work on deception detection.

3) The technical novelty is that prompts are not only contextual tokens but active routing agents: they guide the selection and weighting of experts for each modality dynamically, based on self-attention responses. This prompt-aware expert routing design is new in the context of multimodal deception detection and perhaps also in modality-specific mixture-of-experts frameworks.

4) The use of learnable \emph{gating prompts} for both modalities ($p^{(n,m)}_{a,g}$ and $p^{(n,m)}_{v,g}$)
that participate in cross-modal attention is relatively new. This design provides the model with explicit, learnable control
tokens that modulate the interaction strength between the audio and visual streams. Furthermore, the combination of prompt-driven cross-modal interaction with expert-level adaptive fusion is not typical. Most prior works perform attention-based fusion or expert fusion separately, but not jointly. Thus, while each individual building block (MHSA, MoE, and prompts) is standard, the way they are integrated (using prompt tokens to route and gate cross-modal expert interactions) introduces an architectural synergy, though not a fundamentally new algorithmic concept.

5) The MoME module introduces a prompt-aware gating mechanism for multimodal fusion. While the softmax-based weighting of modality experts is standard, deriving expert weights from cross-modal gating prompts adds a layer of semantic adaptivity, linking prompt representations to modality importance. This constitutes a modest but conceptually coherent extension of conventional MoE-based fusion. I am not sure if this has already been implemented before.

**Weaknesses:**

1) The notation used in Equations (3)--(5) is somewhat inconsistent and could be clarified for better readability.
In particular, $Z^{(n,m)}$ is defined both as the input sequence (Eq. 4) and again as the output of the Transformer block (Eq. 5), which may cause confusion.

It would be clearer to denote the output as $\tilde{Z}^{(n,m)}$ or $Z^{(n+1,m)}$.

Moreover, the description that the Transformer block ``computes feature correlations'' is not entirely accurate, as the block performs contextual representation learning via self-attention rather than explicitly computing correlations. Revising the notation and terminology would improve the technical precision and clarity of this section.

2) The proposed Consistency-Aware Expert Weight Loss aims to regulate the modality weight distribution based on an estimated conflict score \( \kappa \). While the intuition (encouraging uniform weighting when modalities agree and focused weighting
when they conflict) is appealing, but the formulation remains largely heuristic.
The conflict score relies on pairwise cosine dissimilarity between mean modality features, which provides only a coarse
approximation of cross-modal disagreement. Furthermore, the threshold-based division of samples
into ``genuine,'' ``low-conflict,'' and ``high-conflict'' categories introduces manual hyperparameters
and discontinuous supervision. The entropy-based regularization, with fixed target entropies, imposes an arbitrary constraint rather than
deriving from a principled probabilistic framework. Consequently, the loss behaves as a heuristic consistency prior, potentially stabilizing training but offering limited theoretical justification.

3) Some figure/table captions retain notations and references to equations or method details. This makes it difficult to read the paper smoothly, as the reader must frequently return to the methods section. Consequently, I might be missing the experiments asked in the "Questions" part. Please explain them.

4) The paper does not discuss any failing cases, which would be highly informative, especially for behavioral scientists interested in understanding the model's predictions in challenging situations. Additionally, further analysis, such as feature or modality-level ablations in specific cases (datasets have several metadata to relate to) or ad hoc investigation of model decisions, is missing.

5) The limitations of the proposed approach are not clearly articulated, which makes it difficult to assess the boundaries of its applicability.

6) The authors claim several things about deception e.g., L50-L56. However, such statements are not supported by any citations. It is recommended to provide appropriate references to empirical studies or reviews.

7) The authors’ claim regarding multimodal fusion techniques is clear and supported by citations. However, it is worth noting that many cues used for deception and emotion detection overlap considerably. This suggests that the deception detection task may still be technically underdeveloped, possibly due to limited dataset sizes. On the other hand, in affective computing, methods such as Mixture of Modality and Cue Experts (and its variations) have been applied, allowing for more nuanced and fair comparisons. It would strengthen the paper if the authors discussed why such approaches were not considered for deception detection. Some technically relevant papers are as follows:
[A] A Mixture-of-Experts Model for Multimodal Emotion Recognition in Conversations
[B] EMOE:Modality-Specific Enhanced Dynamic Emotion Experts
[C] Discrepancy-Aware Contrastive Learning with Mixture of Experts for Cross-Modal Image-text Semantic Alignment
[D] Hierarchical MoE: Continuous Multimodal Emotion Recognition with Incomplete and Asynchronous Inputs
[E] MMOE:Enhancing Multimodal Models with Mixtures of Multimodal Interaction Experts

8) An Important citation and comparison is missing [F]. In some cases, [F] performs better  than the proposed method and this has to be discussed and included into the relevant tables.
[F] Zhu, Dongliang, et al. "Detecting Deceptive Behavior via Learning Relation-Aware Visual Representations." IEEE Transactions on Information Forensics and Security (2025).

**Questions:**

1) Which is the experiment you remove the interaction branch (cross-modal attention between audio and visual), keeping only unimodal features?

2) To identify which type of prompt contributes most to routing and representation, are testing the effect of different prompts such as:
- Only expert prompts
- Only intra-modality prompts
- Only cross-modality prompts

3) To quantify the benefit of prompt-based instance-adaptive weighting, do you replace prompt-driven routing with either:
- Learned static weights per modality/expert
- Uniform weights (no routing) ?

---

### Official Review · Reviewer_dhiE · 2025-10-29

**Soundness:** 2
**Presentation:** 3
**Contribution:** 2
**Rating:** 4
**Confidence:** 3

**Summary:**

This paper introduces MOMCE, the Mixture of Modality and Cue Experts, an intricate structured model for addressing the challenges in multimodal audio-visual deception detection. The main idea dependes on two-tiered Mixture-of-Experts, 1) Cue Experts for improving the model’s capacity and 2) Modality Experts for utilizing a dynamic gating mechanism to fuse the features derived from the auditory, visual, and synthesized interaction streams. The authors indicate that this approach shows greater performance and robustness than prior fusion approaches.

**Strengths:**

S1) The authors utilizes the principled decomposition of the multimodal fusion issue into two stages.

S2) The inclusion of an Interaction Experts pathway, dedicated solely to synthesizing cross-modal relationships before the final fusion step, can provide a robust intermediate representation that stabilize the modality conflict resolution process.

**Weaknesses:**

W1) The complexity and hierarchical nature of the MoE model occur practical and theoretical weaknesses. Because they dependes on the multiple experts, it is required to make exponential computational resources during both training and inference. However, the paper lacks a comprehensive analysis of this overhead

W2) MoE models are highly susceptible to the expert collapse issue, particularly in domains where data scarcity is common; if the deception datasets used are not sufficiently large or diverse, the individual experts may not specialize properly, leading to increased training instability or poor generalization to out-of-distribution subjects.

W3) Depending on the expert routing decisions for generating the computed gating prompts requires rigorous demonstration of the prompt’s quality and noise immunity.

**Questions:**

Q1) Can we know a more detailed information of the Cue Expert specialization? Do the individual experts mainly learn learn to specialize in certain types of cues, or do they cluster subjects into groups based on their individual deception styles?

Q2) Considering the nature of its application, were any specific experiments conducted to examine the model's fairness and bias mitigation, particularly to make sure that the modality experts do not downweight cues from certain demographic groups when conflict is detected?

Q3) What kinds of regulazation approaches were utilized for uniform expert examination (or excluding the sparse activation issue in MoE architectures)?

---

### Official Review · Reviewer_SoNs · 2025-10-30

**Soundness:** 3
**Presentation:** 2
**Contribution:** 2
**Rating:** 2
**Confidence:** 4

**Summary:**

While the paper proposes a novel two-level mixture-of-experts model (MoMCE) and introduces two regularization losses to improve training, it has critical flaws that undermine its core arguments.
Key disadvantages are as follows:
Unclear ablation study design: The specific structure of baseline models without the MoME or MoCE module is not specified, making it impossible to effectively verify the actual contribution of the proposed components.
Insufficient experimental evidence: The overall experimental results fail to convincingly demonstrate that the method specifically addresses modality conflict and cue heterogeneity. Performance improvements seem to stem more from general MoE feature fusion rather than effective solutions to the stated problems.
Weak baseline comparisons: Most selected baselines are outdated, lacking sufficient comparisons with state-of-the-art methods. Thus, the advancement of this work cannot be strongly justified.
Lack of method interpretability: The functions of the three proposed prompts lack empirical validation. Their roles are more like idealized descriptions, with no association with specific, interpretable deception cues (e.g., microexpressions, tone changes). The entire model resembles a data-driven "black box," lacking a reasonable explanation of its working mechanism.
Based on the above, I believe this paper does not meet the acceptance criteria of ICLR.

**Strengths:**

1:Proposed a modality-and-cue-level mixture-of-experts mechanism: The paper presents a two-level mixture-of-experts structure. This design enables the model to adaptively handle inter-sample differences, enhancing its expressive power and robustness.
2:Introduced two regularization loss functions to improve training effectiveness: Tailored for the two experts, these loss functions effectively alleviate overfitting and modality imbalance during training, boosting the model’s generalization ability.

**Weaknesses:**

1:Unclear ablation study design: The ablation study fails to specify the structure of the model without MoME and MoCE. This lack of clarity prevents verifying the actual effectiveness of the proposed method.
2:Insufficient experimental evidence: The overall experimental results and comparisons with baseline models do not demonstrate that the proposed method addresses challenges like modality conflict and cue heterogeneity. Instead, it merely leverages MoE-based modal feature fusion. The use of attention fusion for video and audio features makes it impossible to validate the method’s specific effects.
3:Weak baseline comparisons: Most selected baselines are outdated, failing to provide strong comparative validity.
4:Lack of method interpretability: There is no proof that the three proposed prompts fulfill their claimed functions as described. Their roles appear to be idealized descriptions. For example, the specific mechanisms of deception cues (e.g., microexpressions, tone changes) are not clarified. The model’s "experts" are mostly data-driven, lacking interpretable semantic alignment and verification.

**Questions:**

1:How to verify that each of the three proposed prompts fulfills the functions as described in the paper?
2:Why not use more multimodal models that jointly process video and audio modalities as baseline comparison models?
3:How to demonstrate the model’s advantages in addressing cue conflict and modality conflict?

---

### Official Review · Reviewer_W1XM · 2025-10-30

**Soundness:** 4
**Presentation:** 4
**Contribution:** 3
**Rating:** 6
**Confidence:** 3

**Summary:**

This paper proposes MoMCE, a novel framework for multimodal deception detection that addresses two key challenges: modality conflict and cue heterogeneity. The method introduces a Prompt-aware Mixture of Modality Experts (MoME) to dynamically weight visual, audio, and interaction modalities based on cross-modal consistency, and a Prompt-aware Mixture of Cue Experts (MoCE) to capture diverse, individual-specific deceptive cues within each modality. The model leverages learnable prompts with semantic biases to instantiate experts and introduces two auxiliary losses: a consistency-aware expert weighting loss to regulate modality fusion, and a cue diversity loss to prevent expert collapse. Experiments on three benchmark datasets (DOLOS, Bag-of-Lies, MU3D) show consistent improvements over state-of-the-art methods in both in-domain and cross-domain settings.

**Strengths:**

Well-Motivated and Structured Architecture:

The paper clearly identifies two under-addressed challenges in multimodal deception detection—modality conflict and cue heterogeneity—and designs a principled architecture to tackle both. The decomposition into MoME and MoCE allows for adaptive, sample-specific modeling of both inter- and intra-modal dynamics, which is a significant conceptual advance over static fusion strategies.

Innovative Use of Prompt-Based Routing and Gating:

The integration of learnable prompts as routing and gating mechanisms is both elegant and effective. By using prompts with different semantic biases to instantiate multiple cue-level experts, the model can capture diverse behavioral patterns without requiring handcrafted features. This approach is well-aligned with recent trends in parameter-efficient and modular learning.

Strong and Comprehensive Experimental Validation:

The evaluation is thorough, covering in-domain, cross-domain, and ablation settings across three diverse datasets. The ablation studies (Table 3) clearly demonstrate the contribution of both MoME and MoCE, and the t-SNE visualizations provide qualitative support for improved feature separability. The hyperparameter sensitivity analysis (Appendix A.3) further strengthens claims of robustness.

**Weaknesses:**

While the application of mixture-of-experts and prompt-based routing is effective, the underlying mechanisms (e.g., prompt gating, diversity loss) are adaptations of existing ideas from MoE and multimodal learning literature. The paper could better position its technical novelty by contrasting with recent MoE-based multimodal models more explicitly.

**Questions:**

1. Can the authors provide qualitative or quantitative evidence (e.g., attention maps, cue attribution) showing that different cue experts indeed specialize in distinct types of deceptive behaviors? This would strengthen the claim of expert diversity.

2. The consistency-aware loss uses a fixed threshold cth 0.5 to distinguish low- vs. high-conflict samples. How sensitive is performance to this threshold? Was it tuned per dataset, or is it a global hyperparameter? A brief analysis would improve reproducibility.

---

### Note · Authors · 2025-11-13

I have read and agree with the venue's withdrawal policy on behalf of myself and my co-authors.